# Screening of Bifidobacteria with Probiotic Potential from Healthy Infant Feces by Using 2′-Fucosyllactose

**DOI:** 10.3390/foods12040858

**Published:** 2023-02-17

**Authors:** Gongsheng Zhang, Hui Sun, Zihe Xu, Ze Tan, Lihong Xiao, Mingxue He, Jiaqi Shang, Anna N. Tsapieva, Lili Zhang

**Affiliations:** 1Key Laboratory of Dairy Science, Ministry of Education, College of Food Science, Northeast Agricultural University, Harbin 150030, China; 2Key Laboratory of Bionic Engineering, Ministry of Education, College of Biological and Agricultural Engineering, Jilin University, Changchun 130022, China; 3Department of Molecular Microbiology, FSBSI Institute of Experimental Medicine, Acad. Pavlov Street, 12, 197376 St. Petersburg, Russia

**Keywords:** *Bifidobacterium*, human milk oligosaccharides, 2′-fucosyllactose, probiotic properties, pilus-like structure

## Abstract

Using 2′-fucosyllactose (2′-FL) as the sole carbon source can be an efficient way to screen bifidobacteria with superior probiotic capabilities since 2′-FL is a key element in promoting the growth of intestinal bifidobacteria in newborns. This approach was used in this work to screen eight bifidobacteria strains, including one strain of *Bifidobacterium longum* subsp. *infantis* BI_Y46 and seven strains of *Bifidobacterium bifidum* (BB_Y10, BB_Y30, BB_Y39, BB_S40, BB_H4, BB_H5 and BB_H22). Studies on their probiotic properties showed that BI_Y46 had a unique morphology with pilus-like structure, a high resistance to bile salt stimulation and a potent inhibitory action on *Escherichia coli* ATCC 25922. Similarly, BB_H5 and BB_H22 produced more extracellular polysaccharides and had a higher protein content than other strains. In contrast, BB_Y22 displayed considerable auto-aggregation activity and a high resistance to bile salt stimulation. Interestingly, BB_Y39 with weak self-aggregation ability and acid resistance had very excellent bile salt tolerance, extracellular polysaccharides (EPS) production and bacteriostatic ability. In conclusion, 2′-FL was used as sole carbon source to identify eight bifidobacteria with excellent probiotic properties.

## 1. Introduction

Human Milk Oligosaccharides (HMOs) are the third most abundant nutritional component in human milk. They are distributed in milk in the form of free lactose derivatives or glycoconjugates with proteins and lipids [1]. Although it cannot be directly used by the human body, as nutrients for infants HMOs can enhance the enrichment ability of specific bifidobacteria in the intestinal tract, reduce intestinal pH, prevent the invasion of pathogenic bacteria and improve infant immunity [2,3].

2′-fucosyllactose (2′-FL) is one of the predominant HMOs. Some microorganisms in the large intestine are capable of utilizing 2′-FL, which can withstand digestive processing [4]. 2′-FL also plays a significant role in the immune process. In some disease models, 2′-FL inhibited the adhesion of *Escherichia coli* to intestinal epithelial cells (Caco-2) and present a potent anti-infective effect [5]. Feeding infants powder containing 2′-FL prevented diarrhea caused by *Campylobacter* [6]. Feeding mice with infant powder containing 2′-FL can improve their immunity [7]. Even more, 2′-FL may have biological activity and reduce plasma inflammatory cytokines [2,8].

*Bifidobacterium* was originally found in the feces of breast-fed infants. It is a gram-positive, non-exercise, strictly anaerobic probiotic. Under the microscope, the bacteria are Y-shaped, spoon-shaped, V-shaped, curved, stick-shaped, or rod-shaped [9]. *Bifidobacterium* can not only promote the growth and development of the body, inhibit tumors and aging, but also regulate intestinal dynamic balance, foster intestinal barrier integrity, reduce the symptoms of colitis and enhance immunity [10,11,12].

*Bifidobacterium* in the intestinal tract of mammals can decompose a variety of complex carbohydrates from the intestinal and dietary sources of the host, thus significantly promoting the metabolism of the host [10]. This characteristic not only ensures the colonization of *Bifidobacterium* in the intestinal tract, but also supplies available nutrition to the host and other intestinal microorganisms through the cross-feeding strategy [1,3].

2′-FL promotes the proliferation of some bifidobacteria and is a critical component in maintaining the balance of intestinal microbiology [1]. The capacity of several bifidobacteria to metabolize 2′-FL has been examined in much research. It has been found that *B. bifidum*, *B. infantis*, *B. breve* and some *B. longum* can grow by 2′-FL. Among them, more strains of *B. bifidum* and *B. infantis* use 2′-FL, but less *B. pseudocatenulatum*. By contrasting newborns fed 2′-FL formula and infants breastfed, Daniels found that the 2′-FL formula baby was well tolerated, the absorption curve of the baby fed with 2′-FL was similar to that of breastfed infants and their growth was similar [13].

Although studies have confirmed that 2′-FL in newborn formula milk powder can regulate the intestinal environment of infants, the resources of *Bifidobacterium* using 2′-FL are still limited. At present, the Chinese government has not promulgated legislation to allow HMO to be added to food as a special nutrient. Consumers like the foreign formula milk powder and probiotic goods with HMOs that have hit the home market. There has not been much research in this sector and the items have not been efficiently manufactured.

In this study, 2′-FL was used as the sole carbon source to screen bifidobacteria able to utilize 2′-FL effectively from infant feces and gene sequencing was used to determine their species. To evaluate their potential probiotic characteristics, extracellular polysaccharides (EPS) yield, cell surface protein content, surface hydrophobicity and self-aggregation ability, bile salt tolerance and bacteriostatic ability of supernatant were analyzed. This study provides strain resources for the development of products containing 2′-FL and *Bifidobacterium*.

## 2. Materials and Methods

### 2.1. Isolation of Bifidobacteria

Bifidobacteria were isolated from the fecal samples of breast-fed healthy termed infants in northeastern China. Fecal samples (2 g) were serially diluted and 1 mL of dilution was spread on Raffinose-*Bifidobacterium* (RB) agar plate [14]. RB plates were put in a round bottom vertical anaerobic culture bag (2.5 L, Qingdao Hope Bio-Technology Co., Ltd., Qingdao, China) containing an anaerobic gas bag (Mitsubishi Gas Chemical Co., Tokyo, Japan) at 37 °C for 48–72 h. All isolates were further purified by streak plating and preliminarily identified based on their morphological and staining characteristics (Gram-positive bacilli) and negative catalase reaction (3% *v/v* H_2_O_2_).

### 2.2. Strains and Culture Conditions

*Bifidobacterium longum* subsp. *infantis* M63 (BI_M63) is isolated from commercial ProTectis drops (BioGaia, Stockholm, Sweden). *Bifidobacterium longum* subsp. *infantis* ATCC 15697 (BI_15697) and *Escherichia coli* ATCC 25922 (EC_25922) were purchased from the China Center of Industrial Culture Collection (CCICC, Beijing, China). All the tested bifidobacterial strains were sub-cultured twice in de Man, Rogosa and Sharpe (MRS, Qingdao Hope Bio-Technology Co., Ltd., Qingdao, China) medium with 0.05% (*w*/*v*) L-cysteine (MRSC) anaerobically at 37 °C. EC_25922 was sub-cultured twice in Luria Bertani (LB) broth at 37 °C for 24 h.

### 2.3. Screening of Bifidobacteria with Effective 2′-FL Utilization

Stationary phase bifidobacterial cultures were washed twice with the same volume of sterilized H_2_O and re-suspended in semirefined de Man, Rogosa and Sharpe broth (sMRS) broth [15] without carbohydrate. Cells were inoculated (1%, *v*/*v*) into sMRS broth containing 1% (*w*/*v*) of glucose (Glu), galactose (Gal), 2′-FL (Sharing Technologies Co., Ltd., Shanghai, China), or no carbohydrate. A Ninety-six-well plate was loaded with 200 μL of cell culture covered with 50 μL of mineral oil and incubated at 37 °C anaerobically for 48 h. The changes in the cell growth were monitored at 595 nm (ΔOD595) for 0 h, 24 h and 48 h. A minimum of 3 independent replications were performed for each carbohydrate.

### 2.4. Identification and Molecular Characterization of Isolated Bifidobacteria

Bacterial genomic DNA was extracted by using a bacterial genomic DNA extraction kit Omega D4015-01 (Omega Bio-Tek Company, Norcross, GA, USA). The discrimination amongst the *Bifidobacterium* isolates at strain level was conducted by randomly amplified polymorphic DNA (RAPD)-PCR analysis using the Bif primer S23: 5'-AGTCAGCCAC-3' S228: 5'-GGACGGCGTT-3' and S396 5'-AGGTTGCAGG-3'. PCR products were analyzed by electrophoresis on 1.2 (*w*/*v*) agarose gels, containing ethidium bromide (0.5 µg/mL) in 1 × TAE. DNA fragments were visualized by UV transillumination and photographed. The confirmation of the identity of presumptive *Bifidobacterium* isolates was conducted by PCR using *Bifidobacterium* specific primers (Bif 164F: 5'-GGGTGGTAATGCCGGATG-3'; Pbi R2: 5'-GACCATGCACCACCTGTGAA-3'). PCR products of the isolates were sequenced by Shenggong Biotechnology Company (Shanghai, China). The 16S rDNA sequences were determined by BLAST alignment analysis.

### 2.5. Morphological Observation of Bifidobacterium by SEM

The bacterial cell was prepared as described previously [16]. Briefly, bacteria were centrifuged, washed with sterilized H_2_O and fixed in 2.5% glutaraldehyde. The cell pellet was dehydrated in 0%, 70%, 80% and 90% (*v*/*v*) ethanol. After being sputter-coated with platinum–gold, the cell morphology was observed by a scanning electron microscope (SEM) S-3400N (Hitachi).

### 2.6. Determination of the Exopolysaccharide Yield

Exopolysaccharide (EPS) was prepared according to our previous method [17]. The 10 mL of bacterial solution was centrifuged at 10,000× *g* for 10 min. The supernatant was mixed with 3 times the volume of 95% ethanol and at 4 °C for 16 h. Then the mixture was centrifuged and the precipitated pellet was resolved in the water. The precipitated EPS was dialyzed with a 3.5 kDa dialysis bag, the water was changed every 8 h and dialyzed for 48 h. The yield of EPS was determined by the Alcian Blue according to McKellar’s method [18].

### 2.7. Determination of Bacterial Surface Protein Content

The bacterial surface protein was prepared as previously described [19]. The 10 mL of bifidobacterial cell was centrifuged at 10,000× *g* for 10 min and washed twice using PBS. The cell was resuspended in 2 mL of 5 mol/L LiCl for 1 h (200 rpm/min, 37 °C). After treatment, the cells were removed by centrifugation (10,000× *g* for 10 min) and the bacterial surface protein resolved in the LiCl solution was precipitated with twice the volume of ice acetone and stored at −20 °C overnight. Bacterial surface protein content was harvested by centrifugation and the content was determined according to Bradford’s method [20].

### 2.8. Determination of Auto-Aggregation Ability

Auto-aggregation ability was determined according to Del Re et al.’s method [21]. The bacterial culture was centrifuged at 10,000× *g* for 10 min, washed and re-suspended in PBS buffer and the absorbance of the upper bacterial suspension at 0 h, 2 h, 4 h and 6 h was determined at 595 nm. Aggregation rate (A%) was calculated as formula: Aggregation rate (A%) = [1 − (A_t_/A_0_)] × 100, where A_0_ is the absorbance of 0 h and A_t_ is the absorbance at various times.

### 2.9. Analysis of the of Simulated Gastric Juice and Bile Salt Tolerance

The tolerance of simulated gastric juice was determined according to Zhuang’s method [22]. The bacterial culture was inoculated into the simulated gastric juice (10 g of pepsin in hydrochloric acid, pH 2.0) at a volume ratio of 1:5, mixed and incubated for 30 min and 3 h. After treatment, the survival cell numbers were determined by drop-plate method [23].

The tolerance of bile salt was determined according to Zhang’s method [17]. Briefly, 1 mL of bacterial solution was centrifuged at 10,000× *g* for 5 min and washed twice with sterilized H_2_O. The washed cell was re-suspended in the MRSC broth with 0.2% porcine bile salt (Shanghai Ruiyong Biotechnology Co., Ltd., Shanghai, China) and cultured for 0 min and 30 min at 37 °C. The cell numbers were monitored by using Chen’s drop plate method.

### 2.10. Determination of the Bacteriostatic Activity of Acellular Supernatant

Using EC_25922 as indicator bacteria, the bacteriostatic ability of the bifidobacterial culture supernatant was determined by the Oxford cup [24]. Briefly, bifidobacteria were cultured for 24 h and centrifuged at 10,000× *g* for 5 min. The supernatant was filtered through a 0.22 μm microporous membrane (Millex^®^-GP, Millipore, Bedford, MA, USA) to generate acellular supernatant. EC_25922 culture was centrifuged, washed and diluted by PBS and coated on the LB agar plate. An Oxford cup (inner diameter 8.0 mm) was placed on a plate. Acellular supernatant (200 µL) of *Bifidobacterium* was added to the Oxford Cup and cultured at 37 °C for 48 h. The bacteriostatic zone size of acellular supernatant was measured by vernier caliper.

### 2.11. Statistical Analyses

All experiments were conducted independently three times and the results were expressed as mean ± standard deviation. Statistical significance (*p* < 0.05) was analyzed by Waller-Duncan significance analysis with SPSS software. Different letters were used to show a significant difference (*p* < 0.05).

## 3. Results and Discussion

### 3.1. Screening of Bifidobacteria Able to Utilize of 2′-FL

RB medium is a non-antibiotic selective media for bifidobacteria with raffinose serving as carbon source and casein and yeast extract as nitrogen source. RB agar supports most growth of bifidobacteria and rare non-bifidobacteria. *Bifidobacterium* growing on RB Agar is a yellow colony surrounded by a yellow halo and sedimentation area [14]. In this study, most bifidobacteria grew well on RB medium. Through the preliminary isolation and screening on RB medium, 126 strains of anaerobic gram-positive bacteria were isolated from the fecal samples of healthy term-infants from northeast of China. These bifurcated or rod-shaped isolates observed by Gram staining microscopic examination with negative catalase were tentatively classified as bifidobacteria.

*Bifidobacterium* is one of the largest human gut commensals in breasted infant intestines, some of which are capable of growing with 2′-FL as a sole carbon source [25]. 2′-FL is a critical component of HMOs, which promotes the growth and development of infants. 2′-FL contributes to synapse formation, neuronal transmission and brain growth, which can enhance learning and memory and support brain development [26]. At present, the resources of *Bifidobacterium* strains that can be grown by 2′-FL as carbon source are limited. It has been reported that *Bifidobacterium longum* subsp. *infantis* and *Bifidobacterium bifidum* are excellent 2′-FL utilizers. *B*. *infantis* possess an ATP-binding cassette (ABC) transporter and glycosyl hydrolase (GH) to utilize 2′-FL, whereas *B. bifidum* can secret extracellular GH to use 2′-FL directly, without an ABC transporter. Previous reports show that the utilization capacity of 2′-FL was different even for bifidobacteria within the same species [27]. This study aimed to effectively screen *Bifidobacterium* with an excellent probiotic function, using 2′-FL as unique carbon source.

The consumption capacity of 2′-FL by 126 isolates was compared (Appendix A). The cell growth of 14 top 2′-FL utilizer with OD595 values greater than 0.5 are shown in Figure 1, with BI_15697 and BI_M63 as the control strains. Previous studies showed that BI_15697 was able to metabolize 2′-FL to support cell growth [28] and express fucosidase (Blon_2335) [29]. Thongaram et al. [25] also found BI_M63 grew better than BI_15697 on sMRS broth with 2′-FL as sole carbon source. Thus, both BI_15697 and BI_M63 were used as the control strains to assess the 2′-FL utilization ability and potential probiotic characteristics of the isolates. The control strains BI_15697 and BI_M63 had excellent ability to utilize both glucose and lactose. Except for Y30 and Y33, the glucose and lactose utilization ability of all the isolates were like that of the control strains BI_15697 and BI_M63. The 2′-FL utilization ability of BI_M63 was much higher than that of BI_15697. Among all the screened strains, Y46 had the strongest ability to utilize 2′-FL and the OD595 of 1.19 was much higher than that of BI_M63. Besides, the growth of Y34, Y45, Y30, H5, Y10, H21 and S40 on 2′-FL exceeded that of BI_M63. 

### 3.2. Identification of Bifidobacterium by RAPD and 16S rDNA Pattern Analysis

RAPD has been extensively used in bacterial genotyping. DNA fragments showing polymorphism can be amplified by PCR and observed by agarose gel electrophoresis, to obtain various DNA fingerprints in order to compare varied species. In the present study, both RAPD (Appendix A) and bacterial morphology were used to distinguish the different species. Isolates Y30, Y33, Y34, Y45 and Y46 were all derived from one sample. Y30 and Y33 were both bifurcated, while Y34, Y45 and Y46 were rod-shaped. Combined with RAPD map, Y30 and Y33 were inferred as the same strain and Y34, Y45 and Y46 were the same strain. According to the utilization of 2′-FL, glucose and lactose and strain morphology, Y30 and Y46 were selected for the subsequent experiment. H3, H4 and H5 were all derived from the same sample. H3 and H5 were rod-shaped and yet H4 was bifurcated. Linked to the RAPD map, both H4 and H5 were selected for subsequent tests. H21 and H22 were both derived from one sample and were similar in all aspects. They were both inferred to be one strain and H22 with better 2′-FL was selected for subsequent tests. Both S39 and S40, with short rod shape and derived from one sample, were inferred to be the same strain. However, both Y10 and Y39 were derived from two different samples, respectively, with different shapes, so they were reserved for further identification. In summary, eight bifidobacteria Y10, Y30, Y39, Y46, S40, H4, H5 and H22 were finally determined for biological identification and subsequent test analysis. Eight bifidobacteria were all identified and sequenced using specific primers and BLAST alignment in the NCBI database. The accession numbers at GenBank are OQ383628-OQ383635. Among eight strains, seven were identified as *B. bifidum*, which were named as BB_Y10, BB_Y30, BB_Y39, BB_S40, BB_H4, BB_H5, BB_H22, and one was *B. infantis* BI_Y46. HMOs in human breast milk help support health and shape the gut microbiota, especially certain species of *Bifidobacterium* [30]. *B*. *infantis* and *B. bifidum* are the major 2′-FL utilizers, however not all the bifidobacteria can use 2′-FL. Consistent with the previous study, the most effective 2′-FL utilizers are *B*. *infantis* and *B. bifidum* in this study. 

Bifidobacteria were shown to be more abundant in breastfed newborn intestines, including *B. longum*, *B. bifidum* and *B. breve*. However, the control group fed with formula milk powder had more non-breast-milk bifidobacteria, including *B. adolescentis*, *B. animalis*, *B. pseudolongum* and *B. catenulatum* [31]. Studies showed that the supplementation of 2′-FL into the formula shortened the gap of gut microbiota between breasted and formula-fed infants [4], attenuated the severity necrotizing enterocolitis and decreased inflammatory cytokines in formula-fed infants [8,32]. 

### 3.3. Observation of Bifidobacterium Morphology by SEM

The morphology of eight bifidobacteria observed by SEM are shown in Figure 2. Most of the strains were bifurcated, rod-shaped and randomly organized. In various habitats, bifidobacteria can take on various shapes and they can also differ in length. In this study, most of the bifidobacteria were rod-shaped and there were many kinds of rod-shapes, such as straight rods, curved rods, blunt round rods at both ends or one end, with varying lengths. On the other hand, both BB_Y30 and BB_Y39 were bifurcated. The surface of the most strains was smooth and partly wrinkled. It is possible that the cell has these characteristics by instinct or that the pretreatment caused the cell to dry out and develop surface wrinkles. Even when the same bifidobacteria are grown in similar circumstances, their morphology is still highly diverse and they could not be the same bifidobacteria, which would provide a certain reference to further determine the specific strains.

BI_Y46 presented a pilus-like structure observed with SEM. The bacterium pili is an important factor in promoting bacterial auto-aggregation and adhesion to the host intestinal epithelial cell [33]. The pilus-related genes in *B. breve* UCC2003 and proteins in *B. bifidum* PRL2010 and *B. longum* BBMN68 were reported, which were recognized to modulate bacteria-host interactions and regulate immune activity [26,34]. The pilus-like structure of *B. brevis* UCC 2003 was identified by immunogold electron microscopy [34]; however, *B. longum* BBMN68 expressed the pili related protein but has no pilus-like structure [33]. Thus, the whole genome of BI_Y46 should be sequenced and the pili related genes and physiological benefits of its pili should be developed.

### 3.4. The EPS Yield of Bifidobacteria

*Bifidobacterium* extracellular polysaccharide (EPS) plays an important role in intestinal immune regulation [35]. The application of strains with high EPS production in foods such as yogurt can not only significantly affect the viscosity and texture of the products, making the products more popular with consumers, but also improve the tolerance of microorganisms in the intestinal tract [36]. EPS secreted by *Bifidobacterium* can not only regulate the balance of intestinal flora, but also improve immunity and reduce cholesterol [37]. Figure 3 displays the EPS production of each experimental strain.

The EPS production of BB_Y39, BI_Y46, BB_H5 and BB_H22 was higher than that of 0.40 mg/mL and, except for BB_H4, the EPS yield of other strains was higher than 0.30 mg/mL. The EPS yield of BB_H4 (0.16 mg/mL) was the lowest. Furthermore, the colonies of the screened bifidobacteria on the MRSC solid medium plate were stickiness, which indicated that they had a strong ability to produce EPS. At present, only a limited number of EPS secreted by lactic acid bacteria can be employed as a thickening agent in dairy products, so it is of practical significance to find bifidobacteria with high EPS production.

The high-yield EPS strains BB_Y39, BI_Y46, BB_H5 and BB_H22 isolated and identified in this experiment may have high adhesion, which will be further verified in the future to provide some reference for practical production and use.

### 3.5. Determination of the Bacterial Surface Protein Content

Bacterial surface proteins are beneficial to the colonization of probiotics in the gastrointestinal system [19,33]. The cell surface protein content of all the tested strains is shown in Figure 3B. The bacterial surface protein content of BB_Y10, BB_Y30, BB_H4, BB_H5 and BB_Y39 was range from 0.38 mg/mL to 0.46 mg/mL, much higher than that of control strains BI_15697 (0.22 mg/mL) and BI_M63 (0.27 mg/mL). BI_Y46 with the pilus-like structure produced moderate surface protein content. Further research will be conducted to determine the relationship between the surface protein of each strain and the colonization and adhesion of probiotics to intestinal epithelial cells.

### 3.6. Analysis of Auto-Aggregation Ability

Auto-aggregation has been proved to play a role in promoting the persistence and survival of intestinal bacteria in the host gastrointestinal tract [38]. The ability of *Bifidobacterium* auto-aggregation at different times is shown in Table 1. In general, the auto-aggregation ability of all the tested bacteria was significantly stronger over time.

After 2 h of treatment, the auto-aggregation ability of BB_H22 (36.21%) and BB_Y10 (39.56%) was significantly higher than that of strains BB_H5 (15.31%), BB_Y39 (17.45%), BI_M63 (17.67%), BI_15697 (18.92%) and BB_S40 (19.67%) and the ability of BI_Y46 (30.56%), BB_Y30 (28.36%) and BB_H4 (23.56%) were in the middle of the above groups. At 4 h, the auto-aggregation ability of BB_H22 reached a maximum of 72.86%, increasing by 36.66%, while BB_H5 had significantly climbed to 50.07% and BI_M63 and BB_Y39 had changed little, still by less than 30%.

At 6 h, the auto-aggregation ability of BB_Y10 and BB_H22 was much higher than the other tested strains, while those of BB_H4, BB_H5, BB_Y30 and BI_Y46 were higher than that of the control strains BI_M63 and BI_15697. The ability of BB_S40 was comparable to that of the control BI_M63 and much better than BI_15697. Unfortunately, the auto-aggregation ability of BB_Y30 was lower than all the other isolates and the control strains. Many factors impact the bacterial auto-aggregation, such as surface protein content, pilus-like structure, lipoteichoic acid, flagella, fimbriae and lipopolysaccharides [39]. In this study, strains with higher surface protein, more EPS production and pilus-like structure showed better auto-aggregation ability. More work needs to explore the relationships between cell-surface molecules and auto-aggregation ability.

### 3.7. Analysis of Tolerance of Simulated Gastric Juice and Bile Salt

To reach and settle in the large intestine steadily, *Bifidobacterium* must overcome the difficulties of a significant quantity of oxidation, penetration and stress of bile salt or gastric juice in the digestive tract [40]. Therefore, the tolerance/resistance of eight isolated bifidobacteria and two control strains to gastric juice and bile salts were evaluated. The pH of human gastric juice typically ranges from 0.9 to 3 and a simulated gastric juice of pH 2.0 was chosen to evaluate bifidobacterial gastric acid resistance (Table 2). In addition, the component and concentration of bile salts in porcine bile salts is closer to human gut bile than that of ox gall. The pretest results showed that the same concentration of pig bile salt caused more considerable damage to bifidobacteria than that of bovine bile salt. Thus, 0.2% of porcine bile salts were employed in this study (Figure 4). 

As compared with the control strains BI_M63 and BI_15697, the *Bifidobacterium* numbers before and after 30 min and 3 h of the simulated gastric juice (pH 2.0) treatment were expressed by log (cfu/mL). At the same time, the reduction rate of bacterial numbers after gastric acid stimulation for 3 h was compared. As shown in Table 2, the numbers of BB_Y39 and BI_15697 had a downward trend with processing time. As compared with BB_Y39 and BI_15697, the other strains had stronger tolerance to simulated gastric juice at pH 2.0. After 30 min of simulated gastric juice treatment, the colony count of BI_M63, BB_Y10, BB_Y30, BB_S40, BB_H4 and BB_H22 was not reduced by simulated gastric juice, whereas the cell numbers of BI_Y46 and BB_H5 decreased slightly after 30 min. The colony numbers of BI_Y46 and BB_H5 decreased by about 1.83 × 10^7^ cfu/mL and 4.99 × 10^7^ cfu/mL, which rise to the original level within 3 h. However, numbers of BI_15697 and BB_Y39 declined by 7.46 × 10^7^ cfu/mL and 9.03 × 10^7^ cfu/mL, respectively, after 30 min. After 3 h of treatment, the survival rates of BI_15697 and BB_Y39 were 69.32% and 76.89%, respectively, and all other strains’ colony counts declined to varying degrees, although their survival rates were close to 100%. The numbers of stronger tolerant strains BI_M63, BB_Y30, BB_H4 and BB_H22 increased slightly, which may be because they could utilize their own cell components as nutrients to support growth under acid stress. More work needs to verify this phenomenon.

Under normal physiological conditions, the bile salt residue in the human intestinal tract is about 0.05–2.0% [41]. The decreased numbers of all tested bacteria treated by 0.2% porcine bile salt for 30 min is shown in Figure 4**.** The numbers of all strains were affected to varying degrees. The tolerance of BB_Y10 to bile salts was the strongest with a cell number reduction of 3.34 × 10^6^ cfu/mL. The bile salt tolerance of BI_Y46, BB_H4 and BB_H22 was moderate. However, the tolerance of BI_15697, BB_Y30, BB_Y39 and BB_S40 were weakest and the cell numbers of those strains decreased by more than 1 × 10^8^ cfu/mL.

### 3.8. Evaluation of the Bacteriostatic Ability of Acellular Supernatant

*Bifidobacterium* is able to withstand the invasion of some pathogens in the human intestinal tract, which is of great significance for maintaining body health and protecting intestinal homeostasis [35]. Therefore, the evaluation of the bacteriostatic ability of the strain is particularly important. In this experiment, the quality control strain EC_25922 was selected as an indicator to explore the bacteriostatic effect of *Bifidobacterium* culture supernatant. With sterile MRSC medium as control, Table 3 displays the ability of each experimental strain’s supernatant to inhibit EC_25922.

BI_15697 did not form a bacteriostatic ring, indicating that the supernatant of BI_15697 had no bacteriostatic effect on EC_25922. Surprisingly, the supernatants of all the other tested strains inhibited the indicator bacteria EC_25922 to varying degrees. The bacteriostatic effect of BB_H4 was strong and the diameter of the bacteriostatic ring was 10.67 mm. The supernatant of BB_H5, BB_Y10, BB_H22, BB_Y30 and BB_S40 had strong bacteriostatic activity and the diameter of the bacteriostatic ring was between 11 and 12.5 mm. The supernatant of BI_M63, BB_Y39 and BI_Y46 had more obvious inhibitory effects on EC_25922 and the diameter of the bacteriostatic ring could reach more than 12.5 mm. BB_Y39 had the most obvious bacteriostatic effect of all strains and the diameter of bacteriostatic ring was 13.67 mm. This lays the foundation for the screening of potential bacteriostatic strains. Various species of Bifidobacterium can produce compounds with bacteriostatic activities, including organic (acetic, lactic and butyric) acids, low-molecular-weight compounds and antibacterial peptides (bacteriocins) [42,43]. Which substance acts on the antibacterial agent needs further study and confirmation.

We have preserved BI_Y46 in China for type culture collection (CCTCC M 20221253), in view of its excellent ability in utilizing 2′-FL and GOS, bile salt tolerance and bacteriostatic activity. Moreover, the whole genome sequence has been completed and submitted to the website of the National Center for Biotechnology Information (NCBI). We will compare its sequence with that of ATCC 15697 and other bifidobacteria with fimbriae in future study.

## 4. Conclusions

In conclusion, eight strains of bifidobacteria were isolated from healthy infant fecal samples using 2′-FL as sole carbon source. Seven strains were *B. bifidum* (BB_Y10, BB_Y30, BB_Y39, BB_S40, BB_H4, BB_H5, BB_H22) and one strain was *B. infantis* (BI_Y46). BI_Y46 with pilus-like structure. The isolates BI_Y46, BB_H5, BB_H22 and BB_Y22 show promising probiotic characteristics. Interestingly, BB_Y39 with weak self-aggregation ability and acid resistance had very excellent bile salt tolerance, EPS producing and bacteriostatic ability. This study provides a selective method for tapping probiotic resources. The bifidobacteria screened in this study, especially BI_Y46, need to be studied further because of their excellent characteristics.

## Figures and Tables

**Figure 1 foods-12-00858-f001:**
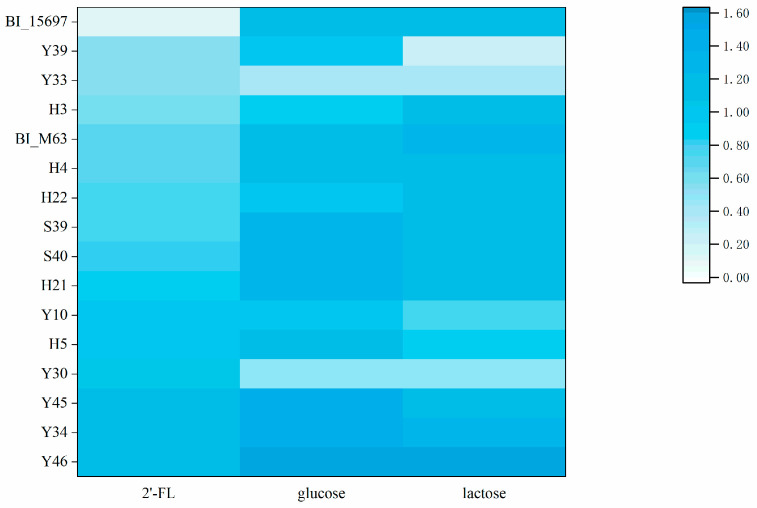
Maximum bacterial growth (change in optical density at 595 nm) in the presence of 1% glucose, lactose and 2′-fucosyllactose (2′-FL) during 48 h of incubation.

**Figure 2 foods-12-00858-f002:**
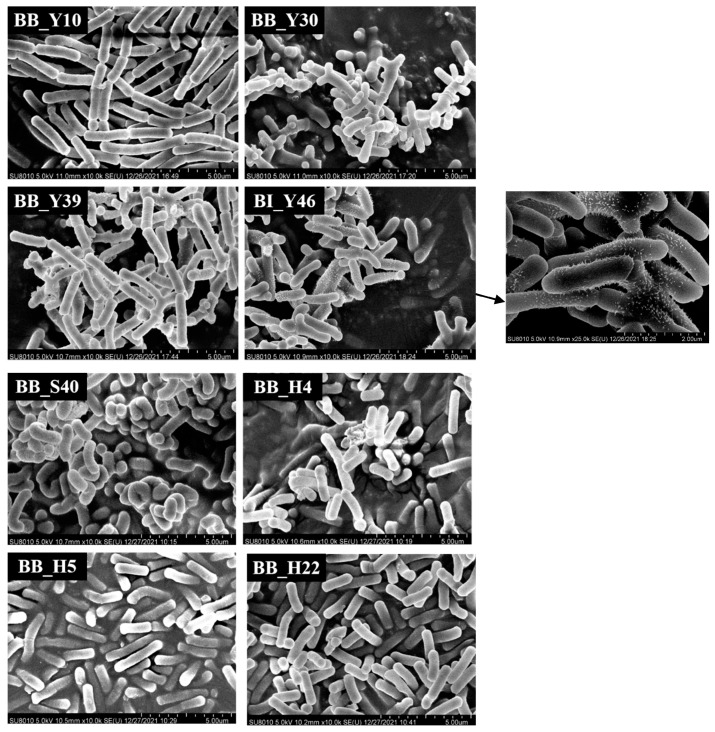
Observation of bifidobacterial morphology using 2′-FL by SEM. BI_Y46 is *Bifidobacterium longum* subsp. *infantis* and the other seven strains are *Bifidobacterium bifidum* (BB_Y10, BB_Y30, BB_Y39, BB_S40, BB_H4, BB_H5, BB_H22).

**Figure 3 foods-12-00858-f003:**
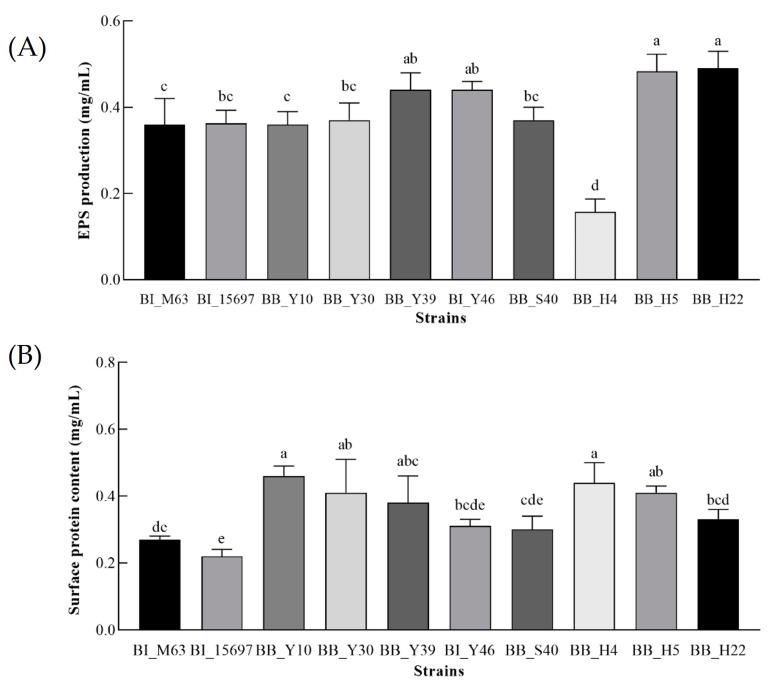
The Exopolysaccharide (EPS) production (**A**) and surface protein contents (**B**) of the screened bifidobacteria. Different lowercase letters represent significant differences among different strains.

**Figure 4 foods-12-00858-f004:**
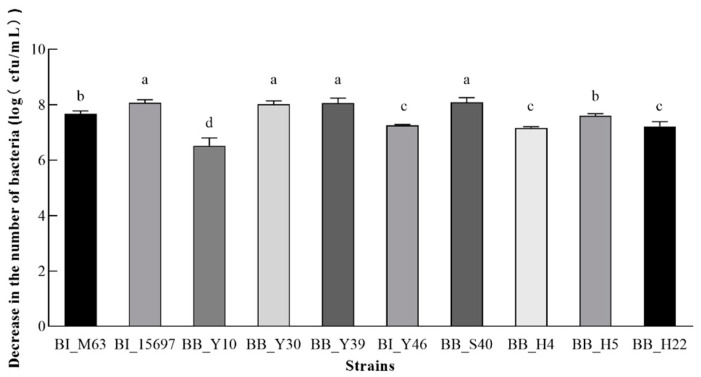
Bile salt tolerance of screened bifidobacteria. The decreased numbers of strains after 30 min of treatment by 0.2% porcine bile salts were used express the bile salt tolerance. Different lowercase letters represent significant differences among different strains.

**Table 1 foods-12-00858-t001:** The auto-aggregation ability of screened bifidobacteria.

Strain	2 h (%)	4 h (%)	6 h (%)
BI_M63	17.67 ± 2.74 ^Cde^	27.37 ± 1.89 ^Bde^	38.98 ± 0.96 ^Ae^
BI_15697	18.92 ± 1.49 ^Cde^	32.80 ± 3.82 ^Bd^	48.43 ± 0.90 ^Ad^
BB_Y10	39.56 ± 3.52 ^Ca^	58.54 ± 5.15 ^Bb^	81.61 ± 3.47 ^Ab^
BB_Y30	28.36 ± 2.75 ^Cbc^	51.31 ± 6.02 ^Bbc^	73.42 ± 1.25 ^Ac^
BB_Y39	17.45 ± 2.24 ^Bde^	23.72 ± 3.35 ^Ae^	30.36 ± 0.29 ^Af^
BI_Y46	30.56 ± 2.51 ^Cbc^	53.31 ± 4.55 ^Bbc^	68.56 ± 3.16 ^Ac^
BB_S40	19.67 ± 0.73 ^Bde^	47.46 ± 2.06 ^Ac^	52.22 ± 2.15 ^Ad^
BB_H4	23.56 ± 2.42 ^Ccd^	52.58 ± 3.44 ^Bbc^	72.88 ± 5.76 ^Ac^
BB_H5	15.31 ± 0.95 ^Ce^	50.07 ± 2.08 ^Bbc^	69.56 ± 5.42 ^Ac^
BB_H22	36.21 ± 3.23 ^Cab^	72.86 ± 3.31 ^Ba^	90.72 ± 2.31 ^Aa^

Note: The result is expressed as mean ± standard deviation (mean ± SD). Different capital letters represent significant difference from the same strain at different times and different lowercase letters show significant difference from different strains at the same time (*p* < 0.05).

**Table 2 foods-12-00858-t002:** Simulated gastric juice (pH 2.0) tolerance of screened bifidobacteria.

Strains	0 h (log (cfu/mL))	30 min (log (cfu/mL))	3 h (log (cfu/mL))	3 h Survival Rate (%)
BI_M63	7.08 ± 0.20 ^Cc^	7.95 ± 0.02 ^Bbc^	8.23 ± 0.15 ^Aa^	116.25 ± 2.58 ^a^
BI_15697	7.88 ± 0.02 ^Aab^	6.09 ± 0.08 ^Be^	5.46 ± 0.28 ^Cf^	69.32 ± 3.56 ^d^
BB_Y10	7.14 ± 0.26 ^Ac^	7.27 ± 0.01 ^Ad^	7.18 ± 0.14 ^Ad^	100.58 ± 2.70 ^b^
BB_Y30	7.81 ± 0.04 ^Aab^	7.85 ± 0.02 ^Abc^	7.80 ± 0.06 ^Abc^	99.34 ± 0.71 ^b^
BB_Y39	7.97 ± 0.02 ^Aa^	6.48 ± 0.07 ^Be^	6.13 ± 0.16 ^Ce^	76.89 ± 2.04 ^c^
BI_Y46	7.88 ± 0.04 ^Aab^	7.76 ± 0.01 ^Bc^	7.81 ± 0.03 ^Abc^	99.06 ± 0.45 ^b^
BB_S40	7.72 ± 0.05 ^Ab^	7.77 ± 0.04 ^Abc^	7.63 ± 0.07 ^Bc^	98.72 ± 0.73 ^b^
BB_H4	7.93 ± 0.02 ^Bab^	8.26 ± 0.07 ^Aab^	7.89 ± 0.04 ^Bb^	99.41 ± 0.63 ^b^
BB_H5	7.99 ± 0.03 ^Aa^	7.68 ± 0.54 ^Bcd^	8.05 ± 0.03 ^Aab^	100.81 ± 0.87 ^b^
BB_H22	8.05 ± 0.02 ^Ba^	8.66 ± 0.53 ^Aa^	7.95 ± 0.03 ^Bb^	98.88 ± 0.35 ^b^

The result is expressed in mean ± standard deviation (mean ± SD). The growth of the same strain was significantly different as shown with different capital letters at different times, and different lowercase letters show significant difference in different strains at the same time (*p* < 0.05).

**Table 3 foods-12-00858-t003:** Inhibitory effect of bifidobacterial supernatant on Escherichia coli ATCC 25922.

Strains	Bacteriostatic Effect
BI_M63	++++
BI_15697	-
BB_Y10	+++
BB_Y30	+++
BB_Y39	++++
BI_Y46	++++
BB_S40	+++
BB_H4	++
BB_H5	+++
BB_H22	+++

Note: The outside diameter of Oxford Cup is 8 mm. -: no bacteriostatic effect; ++: bacteriostatic zone diameter 9.5–11 mm; +++: bacteriostatic zone diameter 11–12.5 mm; ++++: bacteriostatic zone diameter 12.5–14 mm. The experimental results were significant (*p* < 0.05).

## Data Availability

Data is contained within the article.

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
