# Peer review of "Screening of Bifidobacteria with Probiotic Potential from Healthy Infant Feces by Using 2′-Fucosyllactose"

_foods, 2023, doi:10.3390/foods12040858_

Round 1

Reviewer 1 Report

Dear Authors,

Introduction sufficiently introduces to the subject of manuscript. Material, research methods and statistical methods have been properly selected and described. Research results have been clearly presented, and have been pproperly interpreted and discussed in relation to the latest literature. Conclusion in the manuscript is adequate to its aim. The manuscript is scientifically valuable and is recommended for publication in Foods after minor corrections.

I recommend the following:

L 120: In my opinion, the bacterial cell preparation should be described in more details.

L 124, 132, 141, 148 and 153: What were the modifications? (“With minor modification”?)

L 129, 138 and 151: Were modifications made to these methodologies?

The manuscript should be carefully formatted as required in Foods, i.e., use italics for Latin names (L90-91) and in the references, or bold for the year in the list of references. Use subscript (L 87, 99). Use uniform citations for tables and figures (L 302, 394).

Significances refer to the means, not the deviations. Please, place them at the mean value not SD.

Author Response

Thanks for your time and kind comments. Your comments made our manuscript big improvement and closer to be for publication. We read your comments and revised this manuscript very carefully. Hopefully, this version can meet the publishing requirements and be accepted as soon as possible.

Q1: L 120: In my opinion, the bacterial cell preparation should be described in more details.

Response: Thanks for your comments. We have supplemented the morphological observation method in detail. “The bacterial cell was prepared as described previously [16]. Briefly, bacteria were centrifuged, washed with sterilized H2O, and fixed in 2.5% glutaraldehyde. The cell pellet was dehydrated in 0%, 70%, 80%, and 90% (v/v) ethanol. After being sputter-coated with platinum‑gold, the cell morphology was observed by a scanning electron microscope (SEM) S-3400N (Hitachi).”

Q2: L 124, 132, 141, 148 and 153: What were the modifications? (“With minor modification”?) L 129, 138 and 151: Were modifications made to these methodologies?

Response: Thanks for your comments. We changed the sample volume in those method. So, we delete “with minor modification”.

Q3 The manuscript should be carefully formatted as required in Foods, i.e., use italics for Latin names (L90-91) and in the references, or bold for the year in the list of references. Use subscript (L 87, 99). Use uniform citations for tables and figures (L 302, 394).

Response: Thanks for your comments. We checked the manuscript carefully, and italicized Latin names of all bacteria, bolded the years and italicized journal names in the list of references, and used subscript in L87 and 99. All the table and figures were uniformly cited.

Q4: Significances refer to the means, not the deviations. Please, place them at the mean value not SD.

Response: Thanks for your comments. We inserted a space between SD and letters, such as “17.67±2.74 Cde” according to other published papers in Foods.

Reviewer 2 Report

This is an interesting paper in the sense that it proposes an efficient method to screen and select the best probiotic bifidobacteria by using a prebiotic (HMO), and suggests a relevant perspective on the surface protein-colonization/adhesion relationship determination of probiotics. The authors are just invited to consider the following remarks and suggestions:

italicized the scientific names of microorganism genera, species, and subspecies (Lines 89-91, 190-192, 241, 282-289, etc.), and use everywhere Bifidobacterium or B. after it has been written for the first time in the manuscript;

- Line 194: write "...that the utilization capacity of 2'FL..."; 

- Line 237: correct "amony" for "among";

- Lines 301-302: remove "In addition to balancing...lower cholesterol." because of redundancy (previous phrase).

- Add a discussion about the possible action mechanisms of bacteriostatic activity with Bifidobacteria (which compounds are responsible for that?), and cite a reference.         

Author Response

Dear professor,

Thanks for your time and kind comments. Your comments made our manuscript big improvement and closer to be for publication. We read your comments and revised this manuscript very carefully. Hopefully, this version can meet the publishing requirements and be accepted as soon as possible.

Q1: - italicized the scientific names of microorganism genera, species, and subspecies (Lines 89-91, 190-192, 241, 282-289, etc.), and use everywhere Bifidobacterium or B. after it has been written for the first time in the manuscript;

Response: Thanks for your comments. We checked the manuscript carefully, and italicized Latin names of all bacteria, and used B. instead of Bifidobacterium except for the first time in the manuscript.

Q2:- Line 194: write "...that the utilization capacity of 2'FL..."; 

Response: Thanks for your comments. We added “the” in this sentence according to your comment.

Q3:- Line 237: correct "amony" for "among";

Response: Thanks for your comments. We have replaced Amony with Among.

Q4:- Lines 301-302: remove "In addition to balancing...lower cholesterol." because of redundancy (previous phrase).

Response: Thanks for your comments. We deleted this sentence.

Q5 - Add a discussion about the possible action mechanisms of bacteriostatic activity with Bifidobacteria (which compounds are responsible for that?), and cite a reference. 

Response: Thanks for your comments. More discussion and references were supplemented as “Various species of Bifidobacterium can produce compounds with bacteriostatic activities, including organic (acetic, lactic, and butyric) acids, low-molecular-weight compounds, antibacterial peptides (bacteriocins) [42,43]. In this study, which substance acts on anti-bacterial agent needs further study and confirmation.”

  1. Amiri, S.; Mokarram, R.R.; Khiabani, M.S.; Bari, M.R.; Khaledabad, M.A. Characterization of antimicrobial peptides produced by Lactobacillus acidophilus LA-5 and Bifidobacterium lactis BB-12 and their inhibitory effect against foodborne pathogens. LWT 2022, 153, 112449.
  2. Monteiro, C.R.A.V.; do Carmo, M.S.; Melo, B.O.; Alves, M.S.; dos Santos, C.I.; Monteiro, S.G.; Bomfim, M.R.Q.; Fernandes, E.S.; Monteiro-Neto, V. In vitro antimicrobial activity and probiotic potential of Bifidobacterium and Lactobacillus against species of Clostridium. Nutrients 2019, 11, 448.

Reviewer 3 Report

The manuscript describe the species of probiotics Bifidobacterium spp. that demonstrate to be excellent 2'-FL utilizer. 

The approach seem to be not well-finished.  I consider the focus of the study is not suitable for Foods Journal, it should be re-sent to Microorganisms Journal.

Determinations of the initial Bifidobacterium strains using 2'-FL screening data are not provide. 

The heatmap plot is not well explaining the outcome. 

Minor revision:

Is the whasing bacteria with esterile water affecting the integrity of cells?

Italic letter for complete names of bacteria along the manuscript

Amony eight strains, seven were identified - 237

Author Response

Dear professor,

Thanks for your time and kind comments. Your comments made our manuscript big improvement and closer to be for publication. We read your comments and revised this manuscript very carefully. Hopefully, this version can meet the publishing requirements.

All the response to your comment was attached.

Round 2

Reviewer 3 Report

Dear Authors, 

Thanks for the corrections and ammendements. 

PCR products of the isolates were sequenced by Shenggong Biotechnology Company (Shanghai, China). The 16S rDNA sequences were determined by BLAST alignment analysis. 

Have you submmitted the 16S rDNA sequences to NCBI data bases? Could you please provide the number assigned in order to verify that the strains are all belonging to Bifidobacterium.  Please insert this validation data in the manuscript.

Thanks in advance.

Author Response

Response: Thanks for your comments. We have submitted the 16S rDNA sequences to NCBI data bases and the accession numbers at GenBank are OQ383628- OQ383635. All the isolates are Bifidobacterium. In order to show more gene information, we used a new Table 1 to replace the old one.

Sincerely,

Lili Zhang